# Investigating the Effects of Mental Fatigue on Resistance Exercise Performance

**DOI:** 10.3390/ijerph18136794

**Published:** 2021-06-24

**Authors:** Denver M. Y. Brown, Amanda Farias Zuniga, Daanish M. Mulla, Divya Mendonca, Peter J. Keir, Steven R. Bray

**Affiliations:** Department of Kinesiology, McMaster University, 1280 Main St. W, Hamilton, ON L8S 4L8, Canada; fariasza@mcmaster.ca (A.F.Z.); mulladm@mcmaster.ca (D.M.M.); mendonds@mcmaster.ca (D.M.); pjkeir@mcmaster.ca (P.J.K.); sbray@mcmaster.ca (S.R.B.)

**Keywords:** physical performance, electromyography, ego depletion, perceived exertion, resistance training, cognitive control exertion, mental exertion

## Abstract

Mental fatigue can impart negative effects on subsequent physical performance, although the mechanisms underlying these effects are not well understood. This study examined whether mental fatigue confers negative carryover effects on the performance of a set of biceps curls, while also investigating physiological and psychological mechanisms proposed to explain the predicted effect. A randomized, cross-over design was employed. On visit 1, participants (*N* = 10) performed a barbell biceps curl one-repetition maximum (1RM) test. On visits 2–3, participants performed 20 biceps curls at 50% of their 1RM, followed by their respective 10 min experimental manipulation (high vs. low cognitive exertion) and then a second set of biceps curls to exhaustion. Ratings of perceived exertion and electromyography of the biceps brachii, triceps brachii, upper trapezius, thoracic erector spinae and lumbar erector spinae were recorded during the physical task. The total number of repetitions completed was similar across the conditions. Results also failed to show between-condition differences for muscle activation and perceptions of exertion. Future research is needed to build an adequate knowledge base to determine whether there is an effect of mental fatigue on dynamic resistance-based task performance and, if so, identify the mechanisms explaining how and why.

## 1. Introduction

Over the past decade, there has been an emerging interest in the carryover effects of prior cognitive exertion on physical performance [1]. Research initially investigated highly controlled tasks, such as isometric handgrip endurance squeezes [2], but has since evolved to examine physical tasks more commonly engaged in for fitness (e.g., resistance and aerobic exercise) and across competitive sports (e.g., soccer, shooting, cycling) (for a comprehensive overview of the different physical tasks employed, see [3]). This area of inquiry has generally developed along two parallel yet overlapping paths of literature: ego depletion and mental fatigue. Although there are theoretical differences between these phenomena, for simplicity’s sake, we will use the term “mental fatigue” to refer to both phenomena throughout this manuscript (cf. [4]). Recent systematic reviews and meta-analyses have taken an integrative approach to unite these lines of inquiry so that the direction and magnitude of the relationship between mental fatigue and physical performance can be quantified. Evidence indicates mental fatigue confers a negative, small-to-medium sized effect on physical performance [3,5]; however, auxiliary meta-analytic results have suggested these effects may be negligible due to publication bias [6] or mere random chance [7].

While the argument regarding whether or not mental fatigue impacts physical performance has yet to be resolved, Pageaux and Lepers [8] were the first to acknowledge that the type of physical task may play a key role in this relationship. Their systematic review revealed downstream effects of mental fatigue manifest more reliably and with larger effects during tasks that require prolonged submaximal effort regulation (e.g., self-paced exercise, time-to-exhaustion trials) compared to those involving brief maximal effort (e.g., 100 m sprint, long jump). These findings have since been supported by meta-analytic evidence [3]. Of particular interest to the present study is that mental fatigue has been found to have a medium-sized negative effect (*g* = −0.51) on resistance exercise performance [3].

At present, there are considerable knowledge gaps regarding mechanisms underlying the relationship between mental fatigue and physical performance. From a psychological standpoint, research supports the idea that perception of effort is the primary determinant of physical performance at submaximal intensities [9]. Comparatively, much less is known about changes in underlying physiological variables that might explain why perceived exertion increases and performance decreases when people are mentally fatigued. One physiological measure that has demonstrated variability in physical performance attributable to mental fatigue is muscle activity. Two studies found that, after completing an effortful cognitive task designed to induce a state of mental fatigue, muscle activity (measured using electromyography (EMG)) in the forearm flexor muscles during a submaximal isometric handgrip endurance trial was significantly greater compared to a non-fatigued state [2,10]. Because isometric exercise was clamped at 50% of each participant’s maximum voluntary contraction (MVC) for both handgrip tasks (i.e., force generation requirements were the same for both trials), the results indicate it takes greater muscle activation to complete the same amount of work when people are mentally fatigued. Although increased muscle activity has been reported in some studies, not every study has observed these effects. For example, Pageaux et al. [11] observed similar levels of muscle activity across mental fatigue and control conditions when participants performed an isometric leg extension task at 20% MVC, despite physical performance being significantly worse with mental fatigue. However, it is important to recognize the performance demands of the exercise task were lower than the handgrip studies (20% vs. 50%) and involved large muscle groups in the lower extremities versus upper, which invites future studies of boundary conditions where muscle activity effects may and may not be observed.

While several studies have investigated muscle activity during isometric exercise when mentally fatigued, the majority of studies in the mental fatigue–physical performance literature have utilized dynamic exercise protocols. Therefore, EMG recordings during dynamic exercise may also provide important insights into neurophysiological factors that could account for mental fatigue-related increases in perceived exertion and performance decrements. In the only study to investigate effects of mental fatigue on performance and EMG with a dynamic exercise task, Pageaux et al. [12] observed higher activation of the vastus lateralis, but not the rectus femoris, while participants cycled for 6 min at 80% of their peak power output. As with the isometric exercise tasks reported earlier, the demands of the exercise task in the study by Pageaux et al. [12] were standardized (i.e., 80% of peak power output) in the mental fatigue and control conditions, which suggest centrally mediated motor activation patterns were affected by mental fatigue.

Given the aforementioned findings, muscle activity appears to be a promising avenue through which to explore mechanisms that could explain the effects of mental fatigue on perceived exertion and physical performance. However, one shortcoming of research, thus far, is EMG has only been recorded from primary agonist muscles [2,10,11,12]. Thus, it is currently unknown whether mental fatigue may also induce changes in EMG amplitude in other muscles. For instance, because muscles are often recruited in what have been historically referred to as “agonist–antagonist” pairs, it is possible that mental fatigue causes increased co-contraction in antagonist muscles leading to postural rigidity, greater energy expenditure and deleterious effects on performance. In addition to possible effects involving antagonist co-contraction, the motor system responds to increased activation in fatigued muscles and strategically recruits other muscles (i.e., synergists) in order to sustain performance [13]. Therefore, when mentally fatigued, people may consciously, or unconsciously, shift the workload across muscles to continue performing the task.

Changing muscle recruitment strategies may help accomplish physical task goals when fatigued, but this adaptation may come at a cost as movement patterns may also be affected and have implications for injury risk [14,15,16,17]. For example, in response to fatigue, people may move in different ways to increase leverage or create a gravitational advantage, thus altering joint demands [18] and drawing upon contributions from muscles that would not typically be involved in the planned movement. Such aberrant movement patterns or postures, as well as over-activation or overuse of certain muscles induced by fatigue, may increase risk of injuries [19,20]. Accordingly, investigating how different muscles participate in dynamic movements when people are mentally fatigued may have important implications for exercisers and athletes, as well as people who work in occupational settings that require alternating between demanding cognitive and physical tasks.

The purpose of the study was to investigate the effects of mental fatigue on physical performance, perceived exertion and muscle activity from a group of muscles (biceps brachii, triceps brachii, upper trapezius, thoracic erector spinae and lumbar erector spinae) during a submaximal biceps curl resistance exercise task. In line with recent systematic review and meta-analytic findings [3,9], we predicted mentally fatigued participants would complete fewer biceps curl repetitions and report greater perceived exertion compared to controls. Based on prior research [2,10], EMG amplitude in the primary agonist muscles (biceps brachii) was expected to be increased during exercise with mental fatigue. Given the analysis of muscle activity in the additional muscle groups was exploratory, no directional hypotheses were proposed.

## 2. Materials and Methods

### 2.1. Participants

A total of ten active (*M* = 551 ± 356 *SD* min of moderate-to-vigorous physical activity per week) male university students (*M* = 22 ± 3 *SD* years old; *M* = 24 ± 2 *SD* kg/m^2^ BMI), who had at least one year of experience with resistance exercise training (*M* = 45.91 ± 4.55 *SD* kg one-repetition maximum (1RM) biceps curl), participated in this study. A recent review of the literature demonstrated consistent evidence of sex differences in muscle fatigability during dynamic contractions [21], and to avoid this potential confounding factor, we only recruited men. We computed a sample size estimate using G*Power 3.1 [22], which was based on the average (*d* = 1.08) of two large effect sizes (*d* = 0.91 and *d* = 1.25) for changes in resistance exercise performance following exposure to an effortful cognitive exertion manipulation [23]. According to G*Power estimates for a difference between two dependent means (two-tailed), using β = 0.80 and α = 0.01, for our primary outcome measure (total repetitions performed), nine participants were sufficient and we slightly oversampled in case of participant attrition. All participants were screened for contra-indicators of performing vigorous intensity exercise using the Physical Activity Readiness Questionnaire [24] and provided informed consent prior to participation.

### 2.2. Design

The study was a double-blind (experimenter administering physical tasks was blind to the cognitive exertion conditions), within-subject, cross-over design. The order in which participants performed the high- and low-demand cognitive tasks was randomized. The study protocol was reviewed and approved by the McMaster Research Ethics Board (File #2017-132; Approved 7 December 2017).

### 2.3. Measures

#### 2.3.1. Demographics and Physical Activity History

Demographics included self-reported sex and age. Anthropometric data included height and weight, which were obtained using a calibrated weigh scale and tape measure and used to calculate body mass index (mass(kg)/height(m)^2^). The International Physical Activity Questionnaire was used to assess weekly minutes of moderate-to-vigorous physical activity during the past 6 months [25,26].

#### 2.3.2. Physical Performance

The primary dependent variable was the total number of repetitions completed in the set of biceps curls following the cognitive exertion manipulations. For this set, participants were asked to perform as many repetitions as possible until concentric failure.

#### 2.3.3. EMG Amplitude

EMG amplitude was recorded using bipolar surface electrodes from five muscles on the hand-dominant side: biceps brachii, triceps brachii, upper trapezius, thoracic erector spinae and lumbar erector spinae. The skin overlying each muscle was shaved and scrubbed with isopropyl alcohol. EMG signals were differentially amplified (CMRR > 115 dB, input impedance 10 GΩ), band-pass filtered at 10–1000 Hz and sampled at 1000 Hz (AMT-8, Bortec Biomedical Ltd., Calgary, AB, Canada). A 10 s quiet trial and muscle-specific maximal voluntary exertions (MVE) were performed (2 repetitions each with 1 min rest between MVEs) following guidelines from Kendall et al. [27].

#### 2.3.4. Ratings of Perceived Exertion (RPE)

Participants verbally reported their perception of exertion using the Borg CR10 RPE Scale [28]. The scale ranged from 0 (no exertion at all) to 10 (maximum perception of exertion). Participants were asked to provide their RPE at the top of the concentric portion of every biceps curl repetition. Having participants report their RPE was also employed to distract them from counting the number of repetitions competed.

#### 2.3.5. Mental Fatigue

Mental fatigue was measured using a Visual Analogue Scale (VAS; [29]). Participants were instructed thus: “Please mark (X) on the line at the point that you feel represents your perception of your current state of mental fatigue.” The scale consisted of a 100 mm line with anchors ranging from “none at all” on left side of the line corresponding with 0 and “maximal” on the right side of the line corresponding with 100. Scores were calculated by measuring the distance (in mm) to the ‘X’ from the left side of the scale. Mental fatigue was recorded prior to beginning the cognitive exertion manipulations, at 2 min intervals throughout and upon completion.

#### 2.3.6. Task Motivation

Task motivation was assessed before the second set of biceps curls in visits 2 and 3 using a VAS [29]. Participants responded to a single-item question: “For the upcoming exercise task, please mark (X) on the line at the point that you feel represents your perception of your current state of motivation.” The scale consisted of a 100 mm line with anchors ranging from “none at all” on left side of the line corresponding with 0 and “maximal” on the right side of the line corresponding with 100. Scores were calculated by measuring the distance (in mm) from the left side of the scale to where the X was marked.

### 2.4. Experimental Manipulations

The cognitive manipulations were delivered in a 12 min window consisting of five 2 min blocks separated by 30 s breaks in which participants provided ratings of mental fatigue. Each task was delivered on a 17-inch computer monitor.

#### 2.4.1. High Cognitive Exertion

A 10 min computerized version of the incongruent Stroop task [30] was used as the high cognitive exertion manipulation. The Stroop task involves response inhibition, a central component of executive function, and is one of the most commonly used manipulations for examining effects of mental fatigue on physical performance (e.g., [12,31,32]). Using the Presentation software V17 (Neurobehavioral Systems, Inc., Berkeley, CA, USA), the word stimuli (i.e., “black”, “blue”, “green”, “red”, “pink”, “gray”) were presented on a white background in size 48 Times New Roman font. Each 2 min block consisted of 135 trials. Word stimuli were presented for 800 ms, followed by a 100 ms inter-trial interval in which the screen was blank. Participants were instructed to respond as quickly and accurately as possible to each trial by saying aloud the color of the font in which the word was presented and ignore the printed word (e.g., for the word “black” presented in “red,” they would say aloud the word “red”), which required participants to inhibit their dominant response of reading the printed word and instead replace it with the subordinate response of naming the font color.

#### 2.4.2. Low Cognitive Exertion Task

Participants watched a 10 min segment of the documentary *Planet Earth*. Documentary films are commonly used as control tasks in studies investigating the effects of mental fatigue on physical performance (e.g., [33,34,35]), as these manipulations have been found to induce no changes in participants affective state or arousal, ultimately suggesting the task is neither relaxing nor boring [35]. To ensure participants remained engaged while watching the documentary, they were instructed to monitor the content of the narrative script and record instances in which the narrator said the word “water.”

### 2.5. Procedure

An overview of the experimental protocol is presented in Figure 1. Participants completed a total of three visits, with no less than 72 h between each visit to allow for adequate recovery. Participants were instructed to avoid engaging in resistance exercise and consuming caffeinated beverages or foods prior to each session, as well as to get at least 8 h of devoted rest the night before their study session, and confirmed adherence to these requirements prior to each testing session.

Upon arrival at the laboratory for visit 1, participants completed the Physical Activity Readiness Questionnaire, provided informed consent and completed the demographic and physical activity participation questionnaires. Participants then had their height and weight measured. Next, proper technique for performing a biceps curl was demonstrated and participants were fitted with an arm restraint (Northern Lights Arm Curl Blaster, Cornwall, ON) that ensured their arms were kept in a fixed position and shoulder movement was limited, thus, reducing the variability in movement from person to person. Participants then began the protocol to determine their 1RM, starting with a warm-up consisting of five repetitions with only the weight of the bar (7.73 kg). Additional weight was then added to the bar and participants were instructed to perform three repetitions. The weight was gradually increased until they could no longer complete three repetitions. A 3 min rest was given in between each set.

When participants were only able to complete one repetition, that weight was determined to be their 1RM. If participants completed only two repetitions, that weight was determined to be 95% of their 1RM [36] and used to estimate their 1RM. After the 1RM protocol, participants were given a 5 min rest, then performed a familiarization session for the physical task they would perform during visits 2 and 3. For the familiarization set, the weight was set at 50% of their 1RM and participants were instructed to complete as many biceps curls as they could until exhaustion. To standardize the speed at which each biceps curl was performed at, a pre-recorded audio cadence was played aloud and participants were instructed to adhere to the pace by starting the concentric portion of each biceps curl when the audio said “up” and the eccentric portion of each biceps curl when the audio said “down”. Each phase was set at 2 s intervals. Participants were also asked to report their RPE at the top of the concentric portion of each biceps curl repetition.

Visits 2 and 3 followed identical protocols, differing only in the cognitive manipulation performed by participants. Upon arriving at the lab, participants had electrodes attached to their biceps brachii, triceps brachii, upper trapezius, thoracic erector spinae and lumbar erector spinae muscles on the dominant side of their body. Two MVEs were then performed for each muscle. Next, participants were fitted with the arm restraint, completed a warm-up set of 10 curls using only the bar (7.73 kg) and after a 3 min rest, then completed their first set of biceps curls at 50% of their 1RM, in time with the audio cadence. Participants were instructed to aim for 20 repetitions and were stopped by the experimenter once they had completed the 20 repetitions; if participants were not able to complete the 20 repetitions in set 1 of visit 2, to keep both visits consistent, they were stopped at the same number of repetitions in set 1 of visit 3. At the top of the concentric phase of each biceps curl repetition, participants reported their RPE. After the set of biceps curls was completed, participants were led to a separate room and completed the cognitive manipulation task with a different experimenter. Prior to the beginning the cognitive task, participants rated their level of mental fatigue and then reported their level of mental fatigue again at 2 min intervals throughout the task. Upon completion of the cognitive manipulation, participants provided a final rating of mental fatigue and completed the task motivation measure before being led back into the training room to perform the second set of biceps curls under the same parameters as the first set (50% of 1RM, in time with the audio cadence, reporting RPE for each repetition). For this set, however, participants were instructed to perform as many repetitions as possible until they reached exhaustion or could no longer keep up with the audio cadence. During the entirety of the study, the experimenters only interacted with the participants to collect measures and ensure their safety—any form of motivational encouragement was deliberately withheld at all times. Participants were remunerated upon completion of each laboratory visit ($10 CAD for each session).

### 2.6. Data Analysis

Descriptive statistics were computed for all study variables. The effectiveness of our cognitive manipulations on mental fatigue was evaluated using a linear mixed effects model with Time (0, 2, 4, 6, 8 and 10 min), Condition (high vs. low cognitive exertion) and the Time X Condition interaction were set as fixed effects and Subject was set as a random effect. Separate paired samples (Welch’s) t-tests were computed to assess differences in task motivation and the primary dependent variable, total repetitions performed, following the cognitive manipulations.

To assess changes in the secondary dependent variables, RPE and EMG amplitude, an iso-repetition (iso-rep) analysis, based on an iso-time procedure was used [35]. On a participant-by-participant basis, the trial with the least repetitions completed was identified and the paired data points from the two trials were compared. For example, if a participant completed 10 repetitions following the high cognitive exertion condition, but 12 repetitions following the low cognitive exertion condition, data for only 10 repetitions were examined for both conditions for that participant. As each participant completed a different number of repetitions, to compile data across participants for analysis, each set of repetitions was normalized to the total number of repetitions for each participant and the number of repetitions in the set were split into quartiles at 0% (first repetition), 25th, 50th, 75th percentiles and 100% (last repetition).

Prior to analyzing EMG amplitude, separate linear mixed effects models were first performed to evaluate any baseline differences across sessions (i.e., pre-high cognitive exertion vs. pre-low cognitive exertion) in the EMG amplitude of all muscles during the set of curls performed prior to the cognitive manipulations. No significant differences were found for the biceps brachii, upper trapezius, thoracic erector spinae and lumbar erector spinae muscles; however, a significant difference was found for the triceps brachii and this muscle was omitted from further analysis. For the secondary outcome measures, we computed a series of linear mixed effects models to analyze the effects of Iso-rep (1st rep, 25%, 50%, 75%, last rep) and Condition (high vs. low cognitive exertion) on RPE and EMG amplitude (% of MVE) for each muscle. Iso-rep, Condition and the Iso-rep X Condition interaction were treated as fixed effects, with Subject entered as a random effect. Raw EMG data were debiased using the quiet trial, full-wave rectified, low pass filtered using a dual-pass 2nd order Butterworth filter (f_c_ = 6 Hz) and normalized to the MVEs. Processed EMG data were split by repetitions. The first and last repetition of each set were removed from further analysis. Mean normalized EMG amplitudes were calculated for all muscles by each repetition. Due to issues during data collection, 38 out of 500 quartiles were removed from the EMG amplitude analysis.

All analyses were performed in R (version 3.5.1) and R Studio (version 1.1.463, Boston, MA) using the *stats* [37] and *lme4* packages [38]. For the primary outcome measure, a Cohen’s *d* effect size was computed for inclusion in future meta-analyses. Linear mixed models were employed as they provide several advantages over repeated measures analysis of variance [39]. Significance was set at α = 0.05.

## 3. Results

### 3.1. Mental Fatigue

Findings demonstrated a significant main effect of Time (estimate = 5.75 (*SE* = 0.84), *p* < 0.001) and a significant Time X Condition interaction (estimate = −3.84 (*SE* = 1.20), *p* = 0.002), in which ratings of mental fatigue increased to a greater extent during exposure to the high cognitive exertion condition (Figure 2). The main effect of Condition was not significant (estimate = −1.30 (*SE* = 4.67), *p* = 0.78).

### 3.2. Task Motivation

There was no difference in motivation between the conditions (low cognitive exertion: *M* = 8.1 ± 1.0; high cognitive exertion: *M* = 7.62 ± 1.43) prior to completing the second set of biceps curls, *t*(9) = 1.03, *p* = 0.33.

### 3.3. Primary Outcome Measure

#### Physical Performance

The difference in total repetitions performed after the low and high cognitive exertion manipulations was not significant *t*(9) = 0.44, *p* = 0.67, *d* = 0.04 (Figure 3). Five participants performed fewer repetitions and three performed more repetitions in the mental fatigue condition, respectively, while two participants performed the same number of repetitions in each condition.

### 3.4. Secondary Outcome Measures

#### 3.4.1. RPE

Findings revealed a significant main effect of Time (estimate = 1.94 (*SE* = 0.11), *p* < 0.001) in that RPE increased over the number of repetitions completed as participants progressed towards exhaustion (Figure 4). The main effect of Condition (estimate = −0.27 (*SE* = 0.50), *p* = 0.60) and Time X Condition interaction (estimate = 0.06 (SE = 0.15), *p* = 0.68) did not reach statistical significance.

#### 3.4.2. EMG Amplitude

Overall, results indicate there were no significant main effects of Condition, or Time X Condition interactions (all *p*’s > 0.05). A main effect of Time on EMG amplitude (%MVE) was observed in each muscle (all *p*’s < 0.05). In these muscles, average EMG amplitude significantly increased over the number of repetitions completed (Figure 5A–D).

## 4. Discussion

This study aimed to determine whether exposure to a mentally fatiguing cognitive task confers negative carryover effects on the performance of a set of biceps curls to exhaustion, while also investigating physiological and psychological mechanisms proposed to explain the predicted effect. Contrary to our hypothesis, the total number of biceps curls completed did not differ between the conditions. We also failed to observe condition-based differences in muscle activity and perceptions of exertion. Several potential explanations for why we found null effects for physical performance and underlying mechanisms are discussed, in addition to recommendations for researchers in this area of inquiry with regards to improving the design of future studies.

The results from the present study generally oppose the broader literature showing mental fatigue causes significant impairments in physical performance [3,5,7,8,9]. At first glance, it would be reasonable to posit that the physical task was too brief—lasting roughly 1.5 min on average—for effort regulation to have been impacted by mental fatigue. However, negative carryover effects of mental fatigue have been observed in a study which used similar parameters (i.e., one set of resistance exercise performed until exhaustion, ~25–30 repetitions completed on average) to those employed in the present study [40] It is plausible, however, that total repetitions completed until exhaustion during one set of exercise lacks sensitivity to consistently detect differences between conditions. For instance, detecting small, but significant changes in performance is important in resistance exercise-based sporting contexts, such as CrossFit events, where the difference between podium positions can be less than 1% [41]. Future research is needed to address the current lack of data regarding the reliability and sensitivity of dynamic resistance-based performance measures. Nevertheless, this study represents the first attempt to test whether mental fatigue alters muscle activity amongst several muscles during a dynamic resistance task and there are likely numerous factors at play contributing to our findings. While a set of biceps curls seems to be a simple task because it is so common, biomechanically, it is a complex movement and, thus, individual differences in performance can be expected to exist.

Another possibility for why we observed a null effect for physical performance was that mental fatigue did not exacerbate perceptions of exertion. Perceived exertion is a central variable in the psychobiological model [42,43,44], which is a popular theoretical perspective that has been adopted to interpret this body of literature. The psychobiological model posits that individuals decide to quit exercising or reduce their current level of effort because the effort required to continue at the current intensity exceeds their potential motivation or perceived ability to sustain such a level of effort. Although there are a limited number of published empirical studies examining the impact of mental fatigue on dynamic resistance exercise to infer from, findings suggest a dynamic relationship exists between perceptions of exertion and physical performance. For example, Graham et al. [23] found similar maximal scores for RPE across fatigued and non-fatigued conditions, despite participants in the mentally fatigued condition performing fewer repetitions during the physical task. Their findings suggest mentally fatigued participants reached the peak perception of effort that they were willing to tolerate sooner than participants in the control condition. Moreover, the only other study that found a null effect of mental fatigue on total repetitions completed during a dynamic resistance-based workout also failed to observe increased perceptions of effort in a mentally fatigued state [45]. Collectively, these findings provide support for the psychobiological model in that perception of effort is the “cardinal exercise stopper” for fixed-demand tasks [46], or “cardinal exercise regulator” for variable-demand tasks [1], but if perceptions of effort are unaffected, then hypothesized declines in subsequent physical performance cannot be expected. Moving forward, researchers are encouraged to consider using additional measures such as the Repetitions in Reserve-Based Rating of Perceived Exertion Scale [47] to determine whether the number of reps an individual believes they can perform before reaching failure becomes miscalibrated after exposure to a mentally fatiguing cognitive task.

The present findings also provide important insight regarding whether muscle activity is altered in a mentally fatigued state. Considering our findings in light of previous research [2,10], we would have expected that muscle activity would have been higher following the mental fatigue condition in order to perform a similar number of total repetitions compared to the control condition. However, our results indicate this was not the case, as EMG amplitude in the biceps brachii was similar over the work cycle across the conditions. Given that EMG amplitude was also similar across conditions for each of the potential co-contractor muscles assessed, our findings cast doubt on the possibility that compensatory muscle recruitment reactions may have helped participants achieve equivalent performance in a mentally fatigued state compared to the control condition. Despite initial research demonstrating that muscle activity may be a potential neurophysiological mechanism to explain the deleterious effects of mental fatigue on physical performance [2,10], studies that have used tasks involving more complex movements indicate the current evidence is mixed at best [11,12]. More research is needed to determine whether this effect generalizes beyond highly controlled tasks, such as an isometric handgrip endurance squeeze.

One noteworthy strength of the present study was the use of a double blinding approach. In their meta-analysis, Brown et al. [3] identified risk of bias due to inadequate blinding as a major concern in this area of research. Having one experimenter deliver the cognitive manipulations and another administer the physical task ensured that we controlled for any experimenter language and/or behaviour that may have affected participant’s perceptions and performance. Albeit resource intensive, this methodology represents one opportunity in which researchers can refine their experimental protocols moving forward. Reducing bias through such methods will only help to provide a more accurate representation of the magnitude and direction of the effect of mental fatigue on physical performance.

In contrast to the strengths of the present study, there are also some limitations that should be acknowledged. First, the small sample size is one potential explanation for why the hypothesized difference in total repetitions completed were not observed. Our sample size estimate was based on a previous study, which demonstrated large effects of mental fatigue on subsequent resistance exercise performance (*ds* = 0.91–1.25) [23]. In the time since undertaking this study, however, Brown and colleagues’ [3] meta-analysis has indicated that the effect of mental fatigue on subsequent resistance exercise performance pooled across six studies (published and unpublished) is in the medium-sized range (*g* = −0.56). Recalculating a sample size estimate based on a medium-sized effect suggested a sample size roughly three times larger than the one used in this study was necessary, although it is worth noting that previous studies using within-subject designs have observed negative effects of mental fatigue on aerobic endurance performance with as few as eight participants [48,49]. Despite this limitation, it is nevertheless important to report these findings to avoid publication bias, often referred to as the file drawer effect [50], so that effect sizes reported in future meta-analyses are not inflated. Second, previous work has shown age- and sex-related differences in muscle fatigability [21,51,52], thus, our findings may lack generalizability. With regards to age, existing literature investigating the impact of mental fatigue on resistance exercise performance has focused specifically on young adults [11,23,40,45] and we therefore decided to investigate this age group for the purpose of comparability. A third limitation relates to the amount (7.6%) of missing/removed EMG data. Missing and/or noisy values in the EMG signal can occur due to several reasons (e.g., disconnection of electrodes, artifacts) during data collection, particularly while performing a physical task involving complex movements. Linear mixed models are robust to missing data and were therefore employed to analyze EMG amplitude data to circumvent this issue [53]. Fourth, while EMG amplitude reflects muscle activity, there are complex processes unfolding between the neural drive to the muscle and muscle activity, which may affect the interpretation of the EMG amplitude. Lastly, we did not assess postural changes during the biceps curls task; such movement alterations could potentially result in changes in muscle activity in other muscles that were not assessed. Moving forward, psychology researchers would greatly benefit from engaging in multi-disciplinary collaborations with biomechanics laboratories that employ motion capture systems and force plates. Projects stemming from these collaborations could better answer the question of whether people move in different ways to optimize performance in the face of mental fatigue.

## 5. Conclusions

In conclusion, we observed a null effect of mental fatigue on the subsequent performance of a single set of biceps curls to exhaustion. These findings should be interpreted in parallel with the finding that mental fatigue did not exacerbate perceptions of exertion, a key mechanism known to drive subsequent declines in physical performance. We also failed to observe differences in EMG amplitude between the conditions, which casts doubt on muscle activity as a key neurophysiological mechanism underlying the mental fatigue –physical performance relationship. Additional research is needed to build an adequate knowledge base determining whether there is an effect of mental fatigue on dynamic resistance-based task performance and, if so, identify the mechanisms explaining how and why.

## Figures and Tables

**Figure 1 ijerph-18-06794-f001:**
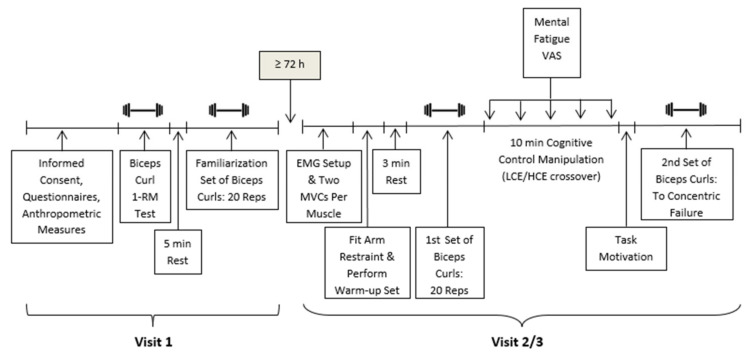
Experimental protocol timeline.

**Figure 2 ijerph-18-06794-f002:**
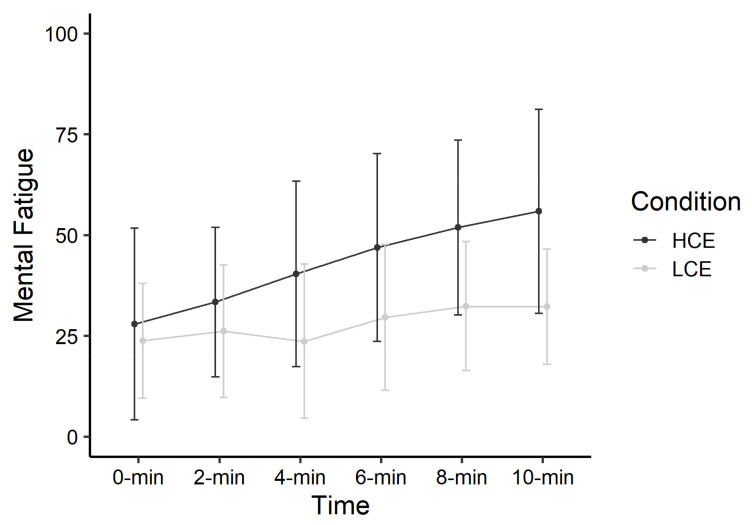
Mental fatigue during cognitive task manipulations, by condition. Data points represent means with standard deviation error bars. LCE = low cognitive exertion; HCE = high cognitive exertion.

**Figure 3 ijerph-18-06794-f003:**
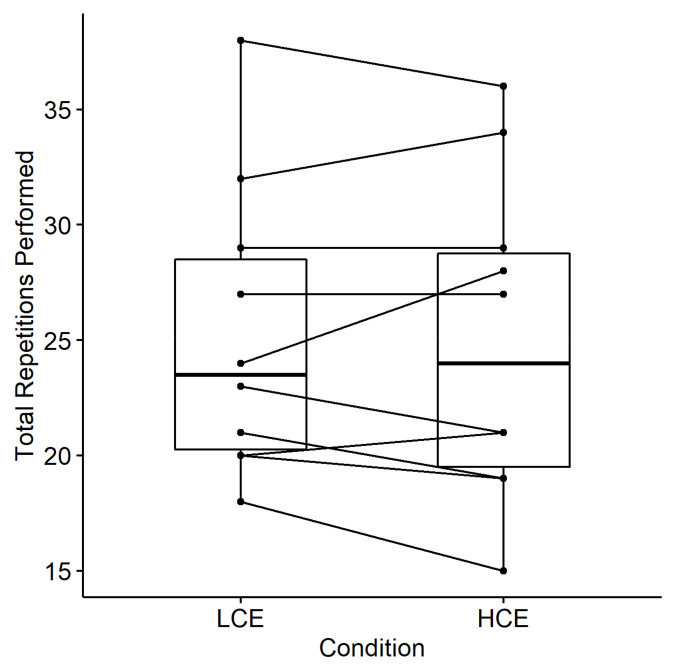
Individual data points and box-and-whisker plots (minimum, median, interquartile range and maximum) for total repetitions performed, by condition. LCE = low cognitive exertion; HCE = high cognitive exertion.

**Figure 4 ijerph-18-06794-f004:**
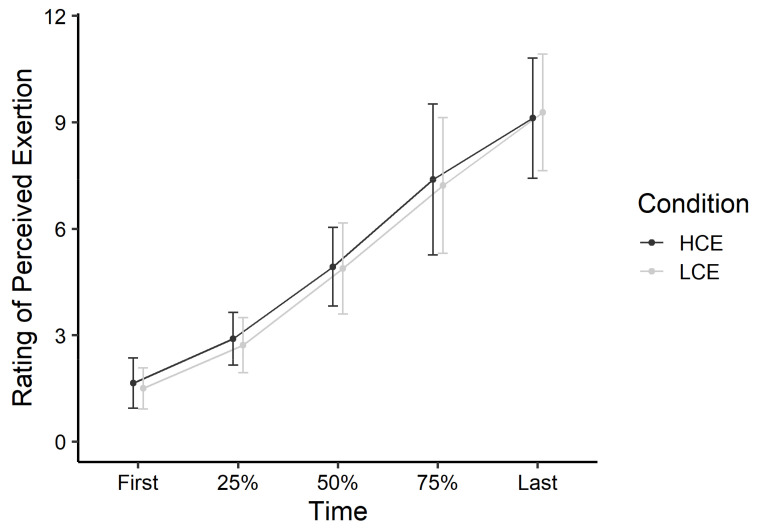
Iso-time ratings of perceived exertion during the physical task, by condition. Data points represent means with standard deviation error bars. LCE = low cognitive exertion; HCE = high cognitive exertion.

**Figure 5 ijerph-18-06794-f005:**
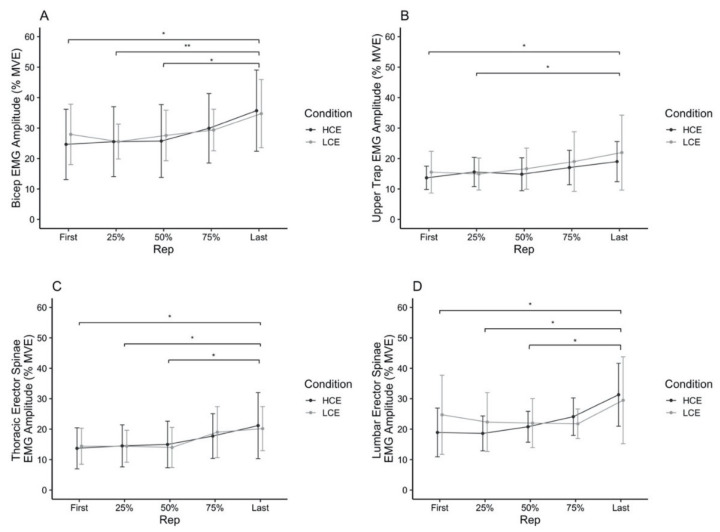
Iso-time average EMG amplitude (% maximum voluntary exhaustion) for the dominant side biceps (**A**), upper trapezius (**B**), thoracic erector spinae (**C**) and lumbar erector spinae (**D**) muscles during the physical task, by condition. Data points represent means with standard deviation error bars. LCE = low cognitive exertion; HCE = high cognitive exertion. Significant comparisons (* *p* < 0.05, ** *p* < 0.001) are shown.

## Data Availability

The data presented in this study are available on request from the corresponding author.

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
