# Peer review of "Investigating the Effects of Mental Fatigue on Resistance Exercise Performance"

_ijerph, 2021, doi:10.3390/ijerph18136794_

Round 1

Reviewer 1 Report

The authors use very current bibliographic citations that reaffirm their statements

Perhaps the sample could have been larger in order to be able to give more support to the study and for it to have greater strength in the research landscape, since each individual does not arrive in the same conditions at each visit to carry out the test, and also that a greater number The smaller the sample size is the uncertainty of the test result.

They do not make explanatory reference about why they use that group with that age range, it could have been any other and the results would be very different, since there are trained subjects in many age ranges. It could be by equipping yourself to previous studies with whom you can make comparisons or discussion.

There is a result described in the introduction that should not go in that part. But in results.

References lower than the year 2000 should be excluded, since it is assumed that there is more up-to-date evidence, although it is true that it is very scarce.

I do not know the sample available to the authors, but the study had to contain a sample of both sexes, not only men, and that the effects produced did not concern only one sex but both.

To search for counter indicators to determine the intensity of exercise, they use the questionnaire from “Thomas, S., Reading, J., & Shephard, R. J. (1992). Revision of the physical activity readiness questionnaire (PAR-Q). Canadian journal of sport sciences”, and it is very old they could have used revisions and adaptations from 5 years ago to date. Idem with the Borg scale.

Author Response

Comment: The authors use very current bibliographic citations that reaffirm their statements

Response: We thank the Reviewer for their positive view on our work.

Comment: Perhaps the sample could have been larger in order to be able to give more support to the study and for it to have greater strength in the research landscape, since each individual does not arrive in the same conditions at each visit to carry out the test, and also that a greater number The smaller the sample size is the uncertainty of the test result.

Response: While we agree that it is possible that every participant comes to the lab in different conditions for each session, they were instructed to follow specific guidelines with regards to avoiding engaging in resistance exercise and consuming caffeinated beverages or foods prior to each session as well as to get at least 8 hr of devoted rest the night before their study session, of which adherence to each was confirmed prior to each testing session. Using a within-subjects design is considered a strength of the present study as each participant serves as their own comparison, whereas in contrast, for between-subject designs, randomization is used with the goal of reducing any potential demographic, physiological and psychological differences between the groups, but this is not always the case.

With regards to sample size, this was acknowledged as a limitation (P12, L462-75), and in hindsight, our power calculation should have been based on a smaller effect size. However, at the time, we based our sample size estimate on effect sizes that had been observed in a study conducted within our lab (Graham et al. 2017).

Comment: They do not make explanatory reference about why they use that group with that age range, it could have been any other and the results would be very different, since there are trained subjects in many age ranges. It could be by equipping yourself to previous studies with whom you can make comparisons or discussion.

Response: Within the mental fatigue – resistance exercise performance literature, studies have focused specifically on young adults. To enable comparability of findings, we elected to recruit this age group. We do, however, recognize that this limits generalizability of our findings as age- and sex differences play a role in muscle fatigability and we have therefore acknowledged this as a limitation (P12, L475-80).

“Second, previous work has shown age- and sex-related differences in muscle fatigability [21,51,52], thus, our findings may lack generalizability. With regards to age, existing literature investigating the impact of mental fatigue on resistance exercise performance has focused specifically on young adults [11,23,40,45], and we therefore decided to investigate this age group for the purpose of comparability.”

Comment: There is a result described in the introduction that should not go in that part. But in results.

Response: After re-reading the introduction, we think we know which section the Reviewer is referring to. We have revised this sentence to clarify that we were not referring to results from the present study but to those from the study of Pageaux et al. (2015) (P2, L88).

“As with the isometric exercise tasks reported earlier, the demands of the exercise task in the study by Pageaux et al. [12] were standardized (i.e., 80% peak power output) in the mental fatigue and control conditions, which suggest centrally-mediated motor activation patterns were affected by mental fatigue.”

Comment: References lower than the year 2000 should be excluded, since it is assumed that there is more up-to-date evidence, although it is true that it is very scarce.

Response: The year in which a study was published does not take away from the quality of the work and preclude us from making reference to those conducted over 20 years ago. To give an example from another field of study, Bollun’s (1989) book on structural equation modeling is considered to be one of the most influential texts in statistics, laying the foundation for current work and as a result is still consistently referred to today. If there is a certain reference that the Reviewer feels there is more up-to-date evidence that should be included instead, we ask that the Reviewer identify the current reference and provide what they view is a more appropriate reference and we would be happy to review the literature to determine whether it is of merit to include it in place of the current reference.

Comment: I do not know the sample available to the authors, but the study had to contain a sample of both sexes, not only men, and that the effects produced did not concern only one sex but both.

Response: Research has established sex differences in muscle fatigability, and we therefore decided to only include males to avoid potential confounders that may influence the results. We have revised the Methods section to highlight why this decision was made (P3, L133-135). This is also acknowledged within the limitations section as noted in our previous comment pertaining to age differences in muscle fatigability.

“A recent review of the literature demonstrated consistent evidence of sex differences in muscle fatigability during dynamic contractions [21], and to avoid this potential confounding factor, we only recruited men.”

Comment: To search for counter indicators to determine the intensity of exercise, they use the questionnaire from “Thomas, S., Reading, J., & Shephard, R. J. (1992). Revision of the physical activity readiness questionnaire (PAR-Q). Canadian journal of sport sciences”, and it is very old they could have used revisions and adaptations from 5 years ago to date. Idem with the Borg scale.

Response: To participate in the present study, interested individuals had to answer no to all of the questions within the PAR-Q. These questions are the same questions included in the first section (“general health questions”) of the more recent PAR-Q+. The major change to the PAR-Q+ is that it is more inclusive of individuals who may answer yes to one of the questions in the first section, in that there are follow-up sections that help to determine whether they may still be ready to exercise despite their condition. Therefore, based on our screening criteria, the recency of the questionnaire did not impact our sampling.

Although more recent iterations of Borg’s perceived exertion scales have been created since the CR10 scale (e.g., CR100), the CR10 scale remains a commonly used measure of perceived exertion with strong psychometric properties (Borg & Borg, 2010).

Reviewer 2 Report

The present study investigated whether mental fatigue induced negative effects on the performance of a set of biceps curls, and whether these effects were also associated to physiological and psychological mechanisms proposed to explain the detrimental effect of mental fatigue. The study is interesting and covers a topic that is worth being investigated. The manuscript is well-written and structured. Introduction and Discussion are clear and complete. I would congratulate the Authors for their work. Here some suggestions and comments that I hope will be useful to improve the scientific quality of the manuscript.

Methods. I would suggest to include a Figure representing the experimental design. This would allow the Readers a better comprehension of the methodological approach and of the measurements employed.

Line 133-134. Why M? Mage should be Age, and so on… I can imagine that M refers to mean, and what about variability? Is standard deviation? Or SEM? Please specify…

Line 135. Please use kg instead of lbs.

Line 159-161. Until concentric failure? Please specify.

Line 172. RPE. RPE has been demonstrated to be affected by mental fatigue. However, have the Authors considered to use the RIR (repetition in reserve) scale? This issue can be expanded also in the Discussion and proposed for possible further studies. Here a reference:

Helms, E.R., Cronin, J., Storey, A., Zourdos, M.C., 2016. Application of the Repetitions in Reserve-Based Rating of Perceived Exertion Scale for Resistance Training. Strength Cond. J. 38, 42–49. https://doi.org/10.1519/SSC.0000000000000218

Moreover, is it possible that reporting RPE for each repetition (at the end of the concentric phase) would induce a mental exertion (that could exacerbate the mental fatigue status)?

Line 202. Were 10 min sufficient to induce a mental fatigue status?

Line 260. Already stated.

Line 385. “Cognitive exertion” instead of “mental fatigue”?

Line 379-387. What about mental fatigue? Did the high cognitive exertion condition really induce mental fatigue? Refer to the mental fatigue paragraph in the results section.

Line 447. “Double blinding approach”?

Author Response

Comment: The present study investigated whether mental fatigue induced negative effects on the performance of a set of biceps curls, and whether these effects were also associated to physiological and psychological mechanisms proposed to explain the detrimental effect of mental fatigue. The study is interesting and covers a topic that is worth being investigated. The manuscript is well-written and structured. Introduction and Discussion are clear and complete. I would congratulate the Authors for their work. Here some suggestions and comments that I hope will be useful to improve the scientific quality of the manuscript.

Response: We thank the Reviewer for their positive comments.

Comment: Methods. I would suggest to include a Figure representing the experimental design. This would allow the Readers a better comprehension of the methodological approach and of the measurements employed.

Response: We have inserted a figure (Fig 1) to represent the experimental protocol timeline.

Comment: Line 133-134. Why M? Mage should be Age, and so on… I can imagine that M refers to mean, and what about variability? Is standard deviation? Or SEM? Please specify…

Response: We have removed “age” and “BMI” as the language following the standard deviation indicates what we are referring to. We have also added “SD” to indicate what the values following ± refer to.

“A total of ten active (M = 551 ± 356 SD mins of moderate-to-vigorous physical activity per week) male university students (M = 22 ± 3 SD years old; M = 24 ± 2 SD kg/m2 BMI), who had at least 1 year of experience with resistance exercise training (M = 45.91 ± 4.55 SD kg  one-repetition maximum [1-RM] biceps curl), participated in this study.”

Comment: Line 135. Please use kg instead of lbs.

Response: We have revised lbs to kg throughout the manuscript.

Comment: Line 159-161. Until concentric failure? Please specify.

Response: We have revised this sentence to indicate we referred to concentric failure (P4, L161-2).

“For this set, participants were asked to perform as many repetitions as possible until concentric failure.”

Comment: Line 172. RPE. RPE has been demonstrated to be affected by mental fatigue. However, have the Authors considered to use the RIR (repetition in reserve) scale? This issue can be expanded also in the Discussion and proposed for possible further studies. Here a reference:

Helms, E.R., Cronin, J., Storey, A., Zourdos, M.C., 2016. Application of the Repetitions in Reserve-Based Rating of Perceived Exertion Scale for Resistance Training. Strength Cond. J. 38, 42–49. https://doi.org/10.1519/SSC.0000000000000218

Response: We were unaware of the Repetitions in Reserve-Based Rating of Perceived Exertion Scale for Resistance Training until now and thank the Reviewer for directing us towards this measure. Considering previous research suggests RPE becomes exacerbated when mentally fatigued, it is possible that the number of reps an individual beliefs they can complete before reaching failure may become miscalibrated and failure may occur before they expect. We have build on our section about RPE to propose using the RIR scale as a possible means to better understand the impact of mental fatigue on resistance exercise performance (P11, L430-4).

“Moving forward, researchers are encouraged to consider using additional measures such as the Repetitions in Reserve-Based Rating of Perceived Exertion Scale [47] to determine whether the number of reps an individual believes they can perform before reaching failure becomes miscalibrated after exposure to a mentally fatiguing cognitive task.”

Comment: Moreover, is it possible that reporting RPE for each repetition (at the end of the concentric phase) would induce a mental exertion (that could exacerbate the mental fatigue status)?

Response: Although reporting RPE likely requires some mental exertion, we are currently unaware of any literature that would suggest that reporting RPE causes problematic levels of mental fatigue. For the most part, participants finished completing their set of biceps curls in less than two minutes, which is well below the threshold that has been shown to lead to mental fatigue induced changes in physical performance (Brown & Bray, 2017). As noted in the Methods section, having participants report RPE served two purposes: 1) to understand their perceived exertion during the task given previous work has shown mental fatigue exacerbates perceptions of exertion, and 2) to distract participants from counting the number of repetitions they performed and thus not allowing them to have a comparison when performing the second session in the crossover trial. We would add further that even if reporting RPE did exacerbate mental fatigue, because participants reported RPE in both the non-fatigue and fatigue trials, these effects on mental fatigue would have been evident in both experimental conditions.

Comment: Line 202. Were 10 min sufficient to induce a mental fatigue status?

Response: At present, there is no established criterion to define whether “mental fatigue” has been successfully induced or not. However, our analysis of the data reported for the mental fatigue visual analogue scale indicates that participants experienced significantly greater levels of subjective mental fatigue when performing the Stroop task compared to watching the documentary. Previous work using the same Stroop task has shown that at least 6 mins of exposure is associated with significant reductions in endurance performance (Brown & Bray, 2017). Moreover, performing the same Stroop task for 10-minutes (Brown & Bray, 2017) has been shown to produce similar levels of mental fatigue to engaging in a moderately demanding continuous performance task (i.e., AX-CPT) for 50-minutes (Brown & Bray, 2019), which has consistently been used in the mental fatigue literature.

Comment: Line 260. Already stated.

Response: After rereading the manuscript, we are unsure of where this was initially stated prior to L260 and kindly ask that the reviewer direct us to repetitive instances of this language.

Comment: Line 385. “Cognitive exertion” instead of “mental fatigue”?

Response: We have revised this sentence and this comment is now not applicable.

“Several potential explanations for why we found null effects for physical performance and underlying mechanisms are discussed, in addition to recommendations for researchers in this area of inquiry with regards to improving the design of future studies.”

Comment: Line 379-387. What about mental fatigue? Did the high cognitive exertion condition really induce mental fatigue? Refer to the mental fatigue paragraph in the results section.

Response: As noted in our earlier response, our analyses indicate that participants were more mentally fatigued following the high cognitive exertion task than after watching the documentary. We have however, revised the language to be more precise about what was examined in the present study (P8, L336-40).

“This study aimed to determine whether exposure to a mentally fatiguing cognitive task confers negative carryover effects on the performance of a set of biceps curls to exhaustion, while also investigating physiological and psychological mechanisms proposed to explain the predicted effect.

Comment: Line 447. “Double blinding approach”?

Response: We have revised this sentence to improve the readability.

“One noteworthy strength of the present study was the use of a double blinding approach.”

References

Hunter, S. K., Critchlow, A., & Enoka, R. M. (2004). Influence of aging on sex differences in muscle fatigability. Journal of Applied Physiology97(5), 1723-1732.

Chan, K. M., Raja, A. J., Strohschein, F. J., & Lechelt, K. (2000). Age-related changes in muscle fatigue resistance in humans. Canadian journal of neurological sciences27(3), 220-228.

Borg G, Borg E. The Borg CR Scales® Folder. Hasselby, Sweden: Borg

Perception; 2010.

Brown, D. M. Y., & Bray, S. R. (2017a). Graded increases in cognitive control exertion reveal a threshold effect on subsequent physical performance. Sport, Exercise, and Performance Psychology, 6, 355–369.

Brown, D. M. Y., & Bray, S. R. (2019). Effects of mental fatigue on exercise intentions and behavior. Annals of Behavioral Medicine, 53, 405–414.

Graham JD, Ginis KM, Bray S. Exertion of self-control increases fatigue, reduces task self-efficacy, and impairs performance of resistance exercise. Sport, Exercise, and Performance Psychology 2017;6:70–88.